# Sri Lankan traditional parboiled rice: A panacea for hyperglycaemia?

**T. P. A. U. Thennakoon**ᵒ, **S. Ekanayake**[ID]*ᵒ

Faculty of Medical Sciences, Department of Biochemistry, University of Sri Jayewardenepura, Nugegoda, Sri Lanka

ᵒ These authors contributed equally to this work.
* sagarikae@sjp.ac.lk, sagarikae@hotmail.com

**Data Availability Statement:** All relevant data are within the paper and its Supporting Information files.

**Funding:** This work was supported by the University of Sri Jayewardenepura, Sri Lanka under grant ASP/01/RE/MED/2016/51 received by SE.

## Abstract

The research aimed to scientifically prove that parboiled Sri Lankan traditional rice elicits lower glycaemic responses comparative to raw unpolished or polished rice. Thus the proximate composition and glycaemic indices (GI) of raw, raw polished, and parboiled traditional Sri Lankan rice (*Oryza sativa L*.) varieties *Godaheenati*, *Batapola el*, *Dik wee*, *Dahanala*, *Unakola samba*, and *Hangimuththan* were studied as comparative data are not available. Cooked parboiled rice contained significantly high moisture (P<0.05) than raw or raw polished. Mineral content was low (<1.5%) regardless of processing. Crude protein was comparatively high (5.8–11.0% DM) with 2.1–5% (DM) fat with raw unpolished and parboiled having higher contents. Digestible starch of raw polished was highest compared to parboiled or raw unpolished (68.8–90.5% DM). Resistant starch was significantly (P<0.05) high in parboiled rice (1.1–7.2%) with the least total dietary fibre in raw polished rice. All varieties of parboiled and raw polished were found to elicit low GI and high GI respectively. High moisture, high resistant starch, and low starch in cooked parboiled rice contributed to low GI compared to raw unpolished or raw polished rice.

## Introduction

Rice (*Oryza sativa* L.) is the staple food of more than half of the World's seven billion population [1]. Daily per capita rice consumption is highest in Asia with an intake of more than 300g (>110 kg per capita annually) while India and China account for 50% of the world's rice consumption [2]. Staple rice is also the major source of carbohydrates and protein for Sri Lankans with a per capita consumption of 114 kg/ year [3]. Sri Lankan adults obtain 72% of their daily energy requirement from carbohydrates which is mainly from rice [4].

Like the world over, Sri Lanka has been witnessing a rapid increase in non-communicable diseases (NCDs) while risk factors such as dysglycaemia, dyslipidaemia, and hypertension have shown a steep incline in the past few decades [5]. A positive correlation between intake of refined raw white rice and risk of type 2 diabetes mellitus is proven with a higher risk in Asians compared to Western populations [6]. Consumption of refined raw white rice contributes largely to the dietary glycaemic load and higher postprandial glucose levels leading to modifiable risk factors and ultimately non-communicable diseases [7]. Consequently, consumption of whole grains or brown rice is recommended in a healthy diet [8]. Therefore, both the

The funders had no role in study design, data collection and analysis, decision to publish, or preparation of the manuscript.

**Competing interests:** The authors have declared that no competing interests exist.

quantity and quality of starch present in frequently consumed carbohydrate-rich foods require prudent consideration.

The glycaemic index (GI) is the blood glucose response to an ingested quantity of carbohydrate in food as compared to the response using a standard reference food [9]. Frequent incorporation of foods with high GI in meals increases the postprandial glucose response and ultimately results in insulin resistance [10]. Epidemiological studies have clearly indicated that low GI diets with higher nutritional value decrease the risk of non-communicable diseases, such as type 2 diabetes, coronary heart disease, etc [11].

During the past decade, a trend in increased consumption of Sri Lankan traditional rice varieties due to their high nutritional and medicinal properties [12,13] is observed. Research has proven that Sri Lankan traditional rice varieties possess high nutritional value when compared to improved varieties [14]. In addition, post-harvest treatment of rice significantly changes the postprandial blood glucose response [15]. Post-harvest processes include milling, parboiling, and storing. Rice milling stage starts with de-husking and ends with polishing to remove bran. Whole grain brown rice retains 100% of its bran and germ [16] and is rich in dietary fibre, vitamins, minerals, and phenolic compounds [17]. Loss of nutrients during the milling process results in an endosperm-rich grain with easily digestible starch [18]. Thus, raw under milled rice compared to raw milled rice has higher nutritional quality [16].

Limited comparative research had been conducted on the effect of different processing methods such as parboiling and polishing on proximate composition and glycaemic indices of Sri Lankan traditional rice. Such information is valuable for consumers, rice cultivators, and nutritionists to formulate rice-based diets and improve the import potential of these varieties. Hence, the objectives of this study were to investigate the effect of parboiling and polishing on nutritional value and glycaemic indices utilizing six Sri Lankan traditional rice varieties.

## Materials and methods

### Materials

Paddy samples of *Godaheenati*, *Batapola el*, *Dik wee*, *Dahanala*, *Unakola samba*, and *Hangimuththan* purchased from the rice preserving centre (*Paramparika Govi Urumayan Rekime Wyaparaya*), Homagama, Sri Lanka were used in the following analyses. Paddy harvested during *Yala* season (2018) were collected into polythene bags, sealed, labelled, and transported on the same day to the laboratory and stored under temperature-controlled conditions until processed.

### Methods

**Sample preparation.** Raw paddy was dehulled (Satake THU 35B), a portion polished (4% polishing level) (Satake TM 05) and the other portion utilized after dehulling. Parboiled rice was obtained by immersing the paddy in boiling water and heating until the paddy grains split open and sun drying as practiced traditionally. The dried parboiled paddy was dehulled but not polished.

Rice samples (dehulled only, polished (4%), parboiled) were washed and cooked (household rice cooker) for 30 to 60 minutes by adding a known amount of water (350–500 mL/250 g rice) as required to reach the cooked texture and taste for each differently processed rice variety. Thus cooked fresh rice was used in the GI study on different days. Proximate analyses were carried out with cooked, oven dried (Memmert, Germany, 55˚C, 3–4 days), milled (IKA ® A11 basic, New Zealand), and sieved (100 mesh sieve) rice flour.

**Moisture and ash.** Moisture of freshly cooked rice grains and rice flour was determined as loss of weight on drying [19]. The weighed samples (0.5000g) were oven dried at 105˚C

until constant weight. The samples were ignited in the muffle furnace (Hobersal, Barcelona, Spain) at 550°C for 5–6 hours for determining the ash content [19].

**Crude protein.**   Crude protein contents were analysed utilizing the [19] micro Kjeldahl apparatus (Paranas Wagner still). A conversion factor of 5.75 was used to determine the protein content.

**Carbohydrates and total dietary fibre.**   Digestible starch and resistant starch contents were determined by amyloglucosidase/ α- amylase (Megazyme total starch assay kit, Ireland). Samples were incubated at 100°C with α-amylase followed by amyloglucosidase at 50°C and starch content assayed as glucose following reaction with glucose oxidase colourimetrically (Spectro UV-VIS Auto/UV-2602, USA). Soluble, insoluble, and total dietary fibre (TDF) contents were determined (Megazyme total dietary fibre assay kit, Ireland) by digesting the samples with α- amylase, protease, and amyloglucosidase.

**Crude fat.**   Samples were digested with 7.7M HCl and fat extracted with petroleum ether and diethyl ether using mojonnier flasks [20].

***In-vivo* glycaemic response.**   Apparently healthy (n = 25, age 20–30 years) volunteers with BMI in the range of 18.5–23 kg/m$^2$ and not under medical treatment were selected for the study. Each volunteer was served the standard twice and each test food once in random order on separate days. Glucoline (gsk Glaxo-Wellcome Ceylon Ltd. Sri Lanka) dextrose monohydrate was used as the standard. The volunteers were served with rice portions (within 1–1.5 hrs following cooking) corresponding to 50 g digestible carbohydrate with gravy made with coconut milk (100 mL), water (125 mL), onions, curry leaves, garlic, fenugreek, turmeric powder, and salt to be ingested within 15 min. Water (250 mL) was also provided to be ingested with the meal/standard [21,22] method.

Finger prick (AccuCheck pricking device) capillary blood samples (100 μL) were collected at fasting and 30–120 min after ingestion of test/ standard food. Blood samples were collected into tubes containing NaF, serum separated, and analysed for glucose within 2 hrs following collection. Blood glucose was estimated with the glucose oxidase enzymatic kit (GOD-PAP, Biolabo, France). Incremental area under the blood glucose curve (IAUC) was determined (using the Trapezoid rule not considering the area below fasting glucose level) [21] for each individual for rice and glucose. GI was calculated as a ratio between the IAUC of the test food and the standard. GI was expressed as the average GI of ten individuals. Glycaemic Load (GL) was estimated by multiplying the GI by the amount of net carbohydrates in each portion [23].

## Statistical analyses

Data were presented as mean ±SD proximate composition and glycaemic index data as mean ±SEM. Data were analysed by SPSS 25.0 statistical software (IBM SPSS Statistics). Descriptive statistics and ANOVA Tukey's posthoc test at 95% confidence interval were used to find the significances.

## Ethical clearance

Ethical clearance for the study was obtained from the Ethical Review Committee of the Faculty of Medical Sciences, University of Sri Jayewardenepura with the reference number 72/17. Informed written consent was obtained from each volunteer prior to the study.

## Results and discussion

### Proximate composition of differently processed rice flour

As comparative data on nutrient content in differently processed traditional Sri Lankan cooked rice is scarce we studied the fate of nutrients when processed differently. The nutrient

**Table 1. Moisture, ash contents, protein and crude fat contents (mean± SD) of differently processed cooked rice flour g/100g in dry weight (fresh weight basis within parenthesis).**

| Variety | Moisture content± SD | | | Ash content± SD | | | Protein content± SD | | | Fat content± SD | | |
|---|---|---|---|---|---|---|---|---|---|---|---|---|
| | Raw | Raw polished | Parboiled | Raw | Raw polished | Parboiled | *Raw | *Raw polished | *Parboiled | Raw | Raw polished | Parboiled |
| *Goda heenati* | 9.8 ±0.3^p | 8.8±0.3^q | 6.5±0.0^r | 1.33 ±0.04^p | 0.76±0.05^q | 1.21±0.09^r | 9.0±0.4^p (3.3 ±0.1) | 7.8±0.6^q (2.8±0.2) | 9.5±0.2^p (2.1±0.0) | 5.0 ±0.2^p (1.7 ±0.1) | 2.6±0.7^q (0.9±0.2) | 3.8±0.3^r (0.8±0.1) |
| *Batapola el* | 5.7 ±0.4^p | 6.4±0.5^p | 5.8±0.1^p | 1.23 ±0.06^p | 0.83±0.02^q | 1.28 ±0.10^p | 9.2±0.4^p (2.9 ±0.1) | 7.4±0.7^q (2.8±0.3) | 8.5±0.3^r (1.5±0.1) | 4.3 ±0.5^p (1.3 ±0.1) | 3.1±0.1^q (1.2±0.1) | 5.0±0.1^p (0.9±0.0) |
| *Dik wee* | 6.9 ±0.2^p | 7.0±0.4^p | 4.9±0.4^q | 1.26 ±0.04^p | 0.84±0.02^q | 0.98±0.06^r | 8.9±0.2^p (2.8 ±0.1) | 6.6±0.6^q (2.3±0.2) | 10.5±0.2^r (1.6±0.0) | 4.0 ±0.3^p (1.3 ±0.1) | 2.7±0.2^q (0.9±0.1) | 4.3±0.3^p (0.7±0.0) |
| *Dahanala* | 6.0 ±0.4^p | 7.0±0.7^p | 5.3±0.1^q | 1.18 ±0.08^p | 0.83±0.04^q | 1.24±0.03^r | 10.5 ±0.2^p (3.7 ±0.1) | 10.1±0.4^p (3.5±0.1) | 11.9±0.4^q (1.9±0.1) | 4.1 ±0.1^p (1.5 ±0.0) | 2.9±0.1^q (1.0±0.0) | 4.0±0.1^p (0.7±0.0) |
| *Unakola samba* | 9.8 ±0.2^p | 9.2±0.3^p | 6.7±0.1^q | 0.81 ±0.06^p | 0.69±0.09^q | 1.03±0.04^r | 8.6±0.2^p (2.7 ±0.1) | 8.9±0.2^p (3.5±0.1) | 9.7±0.7^p (2.0±0.1) | 3.7 ±0.6^p (1.4 ±0.2) | 2.8±0.2^q (1.1±0.1) | 4.8±0.1^r (1.0±0.0) |
| *Hangimuththan* | 8.0 ±0.3^p | 5.2±0.3^q | 5.6±0.1^q | 0.84 ±0.05^p | 0.68±0.03^q | 1.15±0.07^r | 8.3±0.4^p (3.1 ±0.1) | 7.6±0.4^q (2.7±0.1) | 8.3±0.2^p (1.7±0.0) | 4.2 ±0.3^p (1.3 ±0.0) | 2.1±0.0^q (0.7±0.0) | 4.9±0.4^p (1.0±0.1) |

n = 5; *n = 3; SD: Standard Deviation; p, q and r superscripts along a row in each parameter (Moisture, Ash, Protein and Fat) indicate significances among differently processed rice varieties at 95% confidence interval.

compositions of the differently processed 6 traditional rice varieties are presented in Tables 1 and 2. The moisture of cooked rice and rice flour was 61–85% and 4.9–9.8%. The moisture content of cooked rice varied in the order of parboiled > raw > raw polished (Figs 1 and 2). When an edible portion of rice is considered significantly lower (P≤0.05) moisture was present in raw polished rice followed by raw and parboiled rice of each variety. Thus, comparatively the volume of parboiled rice one could consume will contain the highest amount of moisture compared to raw or raw polished rice of each variety. Thus, the available nutrient content of an edible portion of differently processed rice will vary depending on the moisture content of cooked rice. Higher moisture content in cooked parboiled rice could be due to the absorption of water by the starch polymorphs formed during parboiling such as annealed starch, lipid–amylose complexes I and II and, retrograded amylopectin and amylose [24]. This is the first report on moisture contents of differently processed cooked traditional rice of Sri Lanka.

Ash contents of differently processed cooked rice flour ranged from 0.68–1.33% respectively with raw polished varieties having significantly lower (P≤0.05) amounts due to removal of bran compared to raw unpolished and parboiled varieties. The ash contents of milled rice were comparatively lower compared to under-milled rice [25,26]. Similar results have been observed for other under-milled raw traditional Sri Lankan rice varieties [12,27]. Irrespective of processing two white coloured varieties, *Unakola samba* and *Hangimuththan* had lower (P≥0.05) ash contents compared to red varieties. Red pericarp rice varieties contain high content of minerals [28].

**Table 2. Digestible starch, resistant starch and total dietary fibre (mean± SD) contents of differently processed cooked rice flour g/100g in dry weight (fresh weight basis within parenthesis).**

| Variety | Digestible starch content ± SD | | | Resistant starch content± SD | | | Total dietary fibre content ± SD | | |
|---|---|---|---|---|---|---|---|---|---|
| | Raw | Raw polished | Parboiled | Raw | Raw polished | Parboiled | *Raw | *Raw polished | *Parboiled |
| *Godaheenati* | 78.4±2.1ᵃ (25.7±0.7) | 82.1±2.9ᵃ (29.6±1.1) | 76.4±0.4ᵃ (16.6±0.1) | 3.9±0.4ᵖ | 2.5±0.5 q | 6.3±0.2ʳ | 10.8±0.8ˣ | 6.4±0.2ʸ | 11.4±1.0ˣ |
| *Batapola el* | 75.8±3.5ᵃ (23.6±1.1) | 79.0±2.4ᵃ (29.6±0.9) | 73.5±1.6ᵃ (13.1±0.3) | 4.4±0.3ᵖ | 3.3±0.2 q | 6.5±0.7ʳ | 11.1±0.1ˣ | 6.1±0.7ʸ | 11.7±0.3ˣ |
| *Dik wee* | 76.7±0.9ᵃ (24.0±0.3) | 90.5±5.1ᵇ (31.0±1.7) | 74.7±1.4ᵃ (11.6±0.2) | 2.2±0.3ᵖ | 1.1±0.2 q | 6.7±0.2ʳ | 11.6±0.2ˣ | 5.8±0.4ʸ | 11.4±1.0ˣ |
| *Dahanala* | 68.8±3.4ᵃ (23.4±1.1) | 80.7±2.1ᵇ (28.3±0.7) | 73.0±2.5ᵃ (11.9±0.4) | 5.3±0.5ᵖ | 2.1±0.4 q | 7.2±0.5ʳ | 11.6±0.3ˣ | 7.2±0.5ʸ | 10.6±0.4ˣ |
| *Unakola samba* | 78.7±0.9ᵃ (24.7±0.3) | 82.5±1.3ᵇ (32.4±0.5) | 73.6±2.6ᶜ (15.0±0.5) | 5.8±0.6ᵖ | 3.2±0.7 q | 6.7±0.7ʳ | 9.5±0.1ˣ | 6.0±0.1ʸ | 10.5±1.8ˣ |
| *Hangimuththan* | 79.2±1.4ᵃ (24.6±0.4) | 82.2±5.8ᵃ (29.6±2.1) | 76.4±2.1ᵃ (15.2±0.4) | 4.6±0.3ᵖ | 2.7±0.4 q | 6.1±0.4ʳ | 9.1±0.5ˣ | 5.7±0.8ʸ | 10.6±0.4ˣ |

n = 5; *n = 3; SD: Standard Deviation; a, b and c superscripts along a row indicate significances in digestible starch contents, p, q and r superscripts along a row indicate significances in RS contents and x and y superscripts along a row indicate significances in total dietary fibre contents among differently processed cooked varieties at 95% confidence interval.

Crude protein contents of all differently processed cooked rice flour varied in the range of 7–12% (DW) (Table 1) and similar results (7–13% DW) were observed for under-milled traditional raw rice varieties [12,27,28]. Except for *Batapola el*, parboiled rice varieties contained higher protein content (DM) compared to raw and raw polished varieties. Traditional rice varieties contained more protein compared to newly improved varieties [29–31]. However, parboiled rice of all varieties had the least protein content on fresh weight (FW) basis due to the comparatively higher moisture (Fig 1). When considering freshly cooked rice the percentage reduction of protein due to parboiling compared to raw rice ranged between 26–49% (FW) with *Dahanala* showing the highest reduction due mainly to high moisture in cooked rice. However, it was noteworthy that parboiled rice flour contained higher protein than raw or raw polished rice flour where the parboiling process has contributed to retaining more

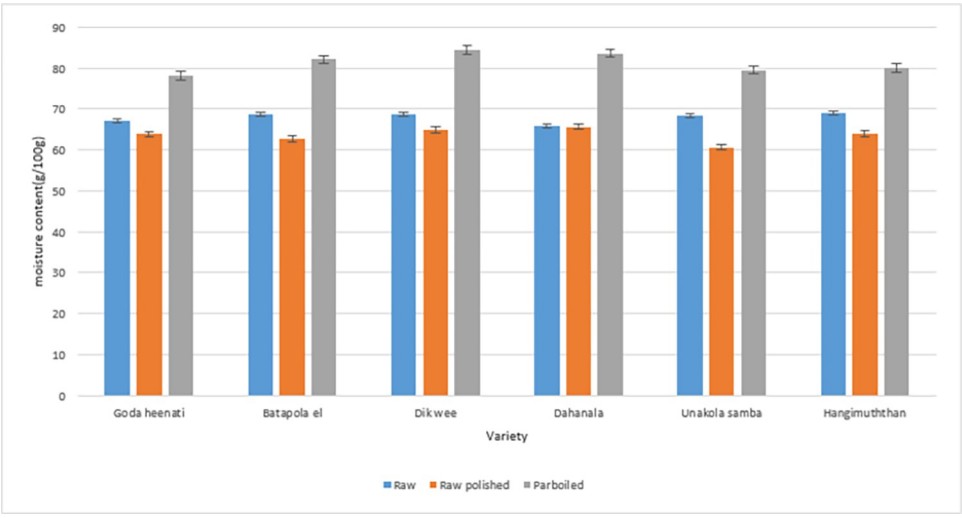

**Fig 1. Moisture contents of differently processed freshly cooked rice (mean±SD).**

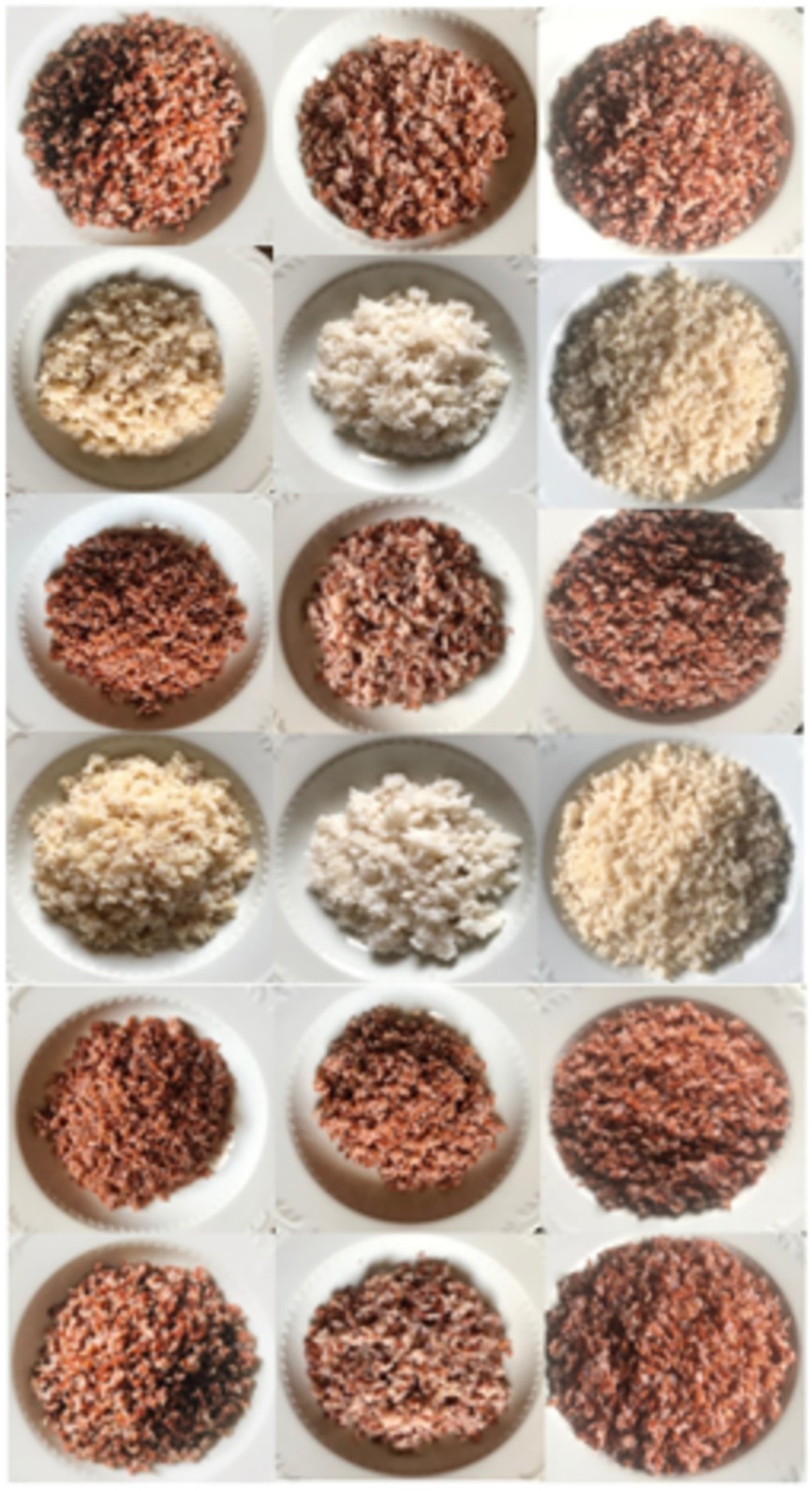

Raw   Raw polished   Parboiled

**Fig 2. Differently processed cooked rice portions containing 50 g of digestible carbohydrate (*Goda heenati*, *Unakola samba*, *Dahanala*, *Hangimuththan*, *Dik wee* and, *Batapola el* respectively).**

protein. Thus incorporation of flour of parboiled rice for food preparation will be more nutritional and healthier.

Crude fat contents of differently processed cooked rice ranged from 2.1–5.0% (DM) (Table 2). Flour of raw polished varieties had significantly lower (P≤0.05) fat contents compared to raw and parboiled varieties due to the removal of the bran layer and germ which are rich in fats whereas under-milled rice and parboiled retains bran and germ which accommodates fat present in the grain [32]. However, the fat content of reported traditional raw rice was less (not exceeding 3.5% DW) when compared to the present study [12,14,27,28]. This could be due to the method of analysis as the lipid content of a given sample varies with the method of analysis [33]. Furthermore, the fat content of all differently processed rice on a fresh weight basis was less than 1.7% with parboiled rice having the least fat content which is due to the high moisture content.

Digestible carbohydrates comprised more than two-thirds of the weight of rice contributing to metabolizable energy. The digestible starch of raw, raw polished, and parboiled rice flour ranged between 68–79%, 79–90.5%, and 73–76% (DW) respectively (Table 2). The majority of raw polished varieties had significantly higher (P≤0.05) digestible starch content due to the removal of outer layers which increased the starch contents of rice grain as only the kernel remains. The digestible starch content of parboiled rice per equivalent amount of raw and raw polished rice on fresh weight was significantly (P≤0.05) less due to higher moisture content. Therefore, comparatively the amount of parboiled rice that one could consume would be less compared to raw or raw polished and thus will contribute to a lower carbohydrate load making these more suitable in the diets of obese and individuals with NCDs.

Resistant starch (RS) contents of differently processed cooked rice flour varied from 1.1–7.2% (Table 2). Cooking increased the RS content in parboiled rice by 6–7 fold, in raw under-milled by 2–5.5 fold, and in raw polished by 2–3 fold compared to uncooked rice. A significant difference in differently processed varieties was observed where raw polished varieties had the least (P≤0.05) and parboiled (P≤0.05) had the highest resistant starch contents. Resistant starch contents of parboiled varieties of the present study were significantly higher confirming the contribution made by retrograded starch (Type 3) to resistant starch due to the parboiling process [34]. This further makes traditional parboiled rice more suitable for the diets of people with NCDs. In addition, the high resistant starch would make parboiled rice an excellent prebiotic food [35].

Total dietary fibre contents of cooked raw, raw polished, and parboiled rice flour varied in the range of 9.1–11.6%, 5.7–7.2%, and 10.5–11.7% respectively (Table 2) where raw polished varieties had significantly lower (P≤0.05) total dietary fibre. Similar data for raw under-milled traditional rice varieties are reported (4.2–6.9% DW) [12,27]. The contribution from insoluble dietary fibre in raw, raw polished, and parboiled varieties was 8.1–9.3%, 5.3–6.2%, and 7.3–9.5% respectively with lower soluble dietary fibre in all differently processed cooked varieties (0.2–3.2%). Similar to total dietary fibre, significantly low (P≤0.05) insoluble and soluble fibre contents were found in raw polished varieties compared to raw or parboiled varieties. As nearly 90% of dietary fibre consists of insoluble dietary fibre (IDF) [36] removal of outer layers could be the reason for the least IDF content in raw polished varieties.

Thus, when considering the nutrients in rice flour, the macronutrient contents were highest in raw unpolished rice followed by parboiled and raw polished rice. Thus consumption of traditional rice is more nutritious as raw unpolished or parboiled rice. The nutrients and dietary

**Table 3. Portion size for GI, peak blood glucose and % peak reduction compared to glucose.**

| Variety | Portion size for GI (g) | | | Peak glucose concentration (mg/dL) | | | % Peak reduction | | |
|---|---|---|---|---|---|---|---|---|---|
| | Raw | Raw polished | Parboiled | Raw | Raw polished | Parboiled | Raw | Raw polished | Parboiled |
| Godaheenati | 185 | 154 | 282 | 115 | 117 | 95 | 6.1 | 4.3 | 28.4 |
| Batapola el | 212 | 158 | 360 | 116 | 120 | 92 | 7.8 | 4.2 | 35.9 |
| Dik wee | 208 | 150 | 412 | 117 | 120 | 94 | 6.8 | 4.2 | 33.0 |
| Dahanala | 224 | 164 | 391 | 117 | 120 | 96 | 6.0 | 3.3 | 29.2 |
| Unakola samba | 202 | 140 | 333 | 115 | 116 | 93 | 8.7 | 7.8 | 34.4 |
| Hangimuththan | 203 | 160 | 311 | 121 | 124 | 112 | 5.8 | 3.2 | 14.3 |

n = 10.

fibre reduced significantly (P< 0.05) irrespective of the lower level of polishing (4%) rice was subjected to in the present study. Thus subjecting rice to higher levels of polishing as available in the market will contribute to increased digestible carbohydrate which is easily available for absorption and rapidly increase glucose and insulin responses.

## Glycaemic responses of differently processed rice

The data related to glycaemic response, glycaemic indices and glycaemic loads of differently processed rice varieties are presented in Tables 3 and 4. We describe the effect of processing on traditional Sri Lankan rice in relation to the glycaemic response and the effect the inherent nutrients have on the said response. This is the first report on such an attempt with Sri Lankan traditional rice to the best of our knowledge.

Portion sizes given for the GI determination varied widely (140g to 412g) (Table 3) where raw polished varieties had the lowest and parboiled varieties had the highest weight (g). The higher moisture content in cooked parboiled rice (Fig 1) made the portion sizes larger (volume) (Fig 2) and according to volunteers this portion was much larger than a normal edible portion.

When the glycaemic curves (Fig 3) were studied the average peak glucose responses were observed at 30–40 minutes following ingestion for all differently processed traditional rice varieties. The incremental area under the curve (IAUC) of all rice varieties were significantly (P≤0.05) lower than glucose. Parboiled rice (1720–2281) had the least area followed by raw varieties (2553–3213) with highest in raw polished varieties (3105–3410). Some parboiled improved and non-parboiled traditional rice produced significantly (P≤0.05) lower IAUC for all rice varieties when compared to standard glucose [37].

**Table 4. Glycaemic indices (mean±SEM) and glycaemic loads of differently processed rice.**

| Processing Variety | Raw | | Raw polished | | Parboiled | |
|---|---|---|---|---|---|---|
| | GI ± SEM | GL | GI ± SEM | GL | GI ± SEM | GL |
| Godaheenati | 58±4 | 29 | 76±5 | 38 | 45±3 | 23 |
| Batapola el | 54±6 | 27 | 76±5 | 38 | 47±5 | 23 |
| Dik wee | 69±7 | 34 | 70±4 | 35 | 43±3 | 22 |
| Dahanala | 60±5 | 30 | 69±3 | 35 | 48±4 | 24 |
| Unakola samba | 65±8 | 33 | 73±5 | 36 | 44±3 | 22 |
| Hangimuththan | 55±7 | 27 | 74±4 | 37 | 43±3 | 22 |

n = 10.

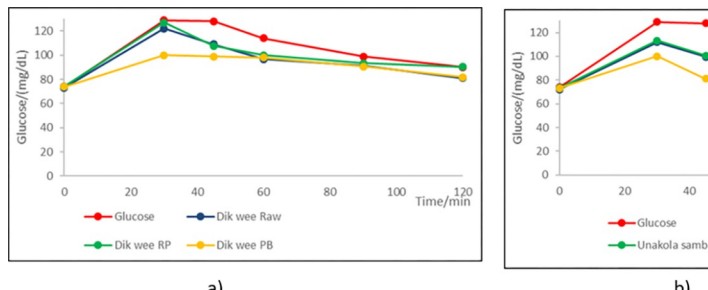

**Fig 3.** Average glucose response of differently processed a) *Dik wee* (red rice variety) over standard glucose b) *Unakola samba* (white rice variety) over standard glucose (RP-raw polished; PB- parboiled).

The average peak glucose concentrations were also highest in raw polished (116–124 mg/dL) followed by raw rice (115–121 mg/dL) and parboiled rice (92–112 mg/dL). In contrast, percentage peak reduction was highest in parboiled rice (14.3–34.4%) and least in raw polished rice varieties (5.8–8.7%) in comparison to glucose. According to Nisanka & Ekanayake [31] parboiled Nadu variety elicited lower mean peak glucose concentration followed by raw samba and basmati variety with the highest mean peak glucose concentration. Both raw and parboiled rice varieties contained high resistant starch and total dietary fibre contents (Table 2) compared to raw polished varieties which have contributed to decreasing the glycaemic indices of raw and parboiled rice.

In accordance with above observations the GI values of raw unpolished, raw polished, and parboiled were between 54–69, 69–76, and 43–76 (Table 4). A clear decline in GI (low) of parboiled varieties was observed. Two of the raw unpolished varieties (*Batapolael* and *Hangimuththan*) also elicited low glycaemic indices ($\leq$55). The remaining raw varieties (*Godaheenati*, *Dik wee*, *Dahanala*, and *Unakola samba*) and one raw polished variety (*Dahanala*) were categorized as medium glycaemic index (56–69). Except for *Dahanala* other raw polished varieties were categorized as high glycaemic index ($\geq$70). The present study data also proved that there is no correlation between the glycaemic index and the colour of the pericarp [30,37]. When considering the effects of inherent nutrients in rice, negative non-significant ($P\geq0.01$) correlations between protein, fat, digestible starch, and total dietary fibre contents present in 50 g carbohydrate containing portions of raw, raw polished, and parboiled rice varieties and their corresponding glycaemic indices were observed. This clearly demonstrated the quantity of inherent nutrients in rice is not adequate to significantly impact the glycaemic index. However, the impact of RS on GI was apparent as a significant negative correlation ($P = 0.000$; $r = -0.858$) was observed with glycaemic index and RS content of rice portion given to determine the GI of differently processed rice of each variety (Fig 4A). Glycaemic load (GL) values of raw, raw polished, and parboiled varieties were in the ranges of 27–34, 35–38, and 22–24 respectively. Thus, the GL for the portions given for glycaemic index determination were categorized as high ($>20$) for all varieties irrespective of processing. A significant positive correlation ($r = 0.932$; $p = 0.000$) was observed between rice portion sizes used to determine the GI and the moisture content in 50g carbohydrate portions (Fig 4B). Therefore, when considering a normal edible portion, GLs of all parboiled varieties which contained high moisture reduced to medium glycaemic load whereas raw polished rice still contained a high glycaemic load.

## Conclusions

It was apparent from the above results, that processing causes significant changes in nutrient content and the glycaemic response of a particular rice variety. Irrespective of variety the

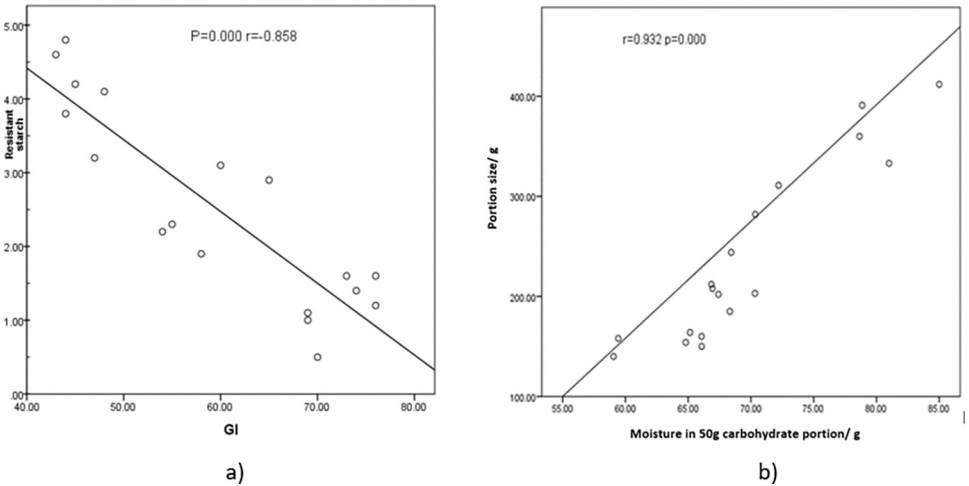

**Fig 4.** Correlation between a) Glycaemic index and resistant starch content in the portion given to study the glycaemic index b) Moisture content in the portion given to study the glycaemic index and the portion size.

moisture content of cooked parboiled was highest which caused to reduce the nutrients but not significantly when compared to raw rice. In contrast, polished rice contained mainly digestible starch with the least amount of protein, fat, dietary fibre, and resistant starch. Parboiled rice elicited lower GI compared to raw and polished rice. The GL of parboiled reduced significantly when an edible portion was considered as the volume of the portion that could be ingested became low due to high moisture retention in parboiled rice. Thus parboiled under milled traditional rice irrespective of variety was nutritionally and health-wise more superior for consumption compared to raw or raw polished rice. In addition, flour of parboiled rice with high protein, fat, resistant starch, and low carbohydrate would be most suitable for use in the food industry as a functional ingredient.

## Supporting information

**S1 File.**
(XLSX)

## Acknowledgments

Volunteers who participated in the glycaemic index study and technical support extended by Ms Ganeshika Amarathunga and Mr GL Chathuranga are acknowledged.

## Author Contributions

**Conceptualization:** S. Ekanayake.

**Data curation:** T. P. A. U. Thennakoon, S. Ekanayake.

**Formal analysis:** T. P. A. U. Thennakoon, S. Ekanayake.

**Funding acquisition:** S. Ekanayake.

**Investigation:** T. P. A. U. Thennakoon.

**Methodology:** T. P. A. U. Thennakoon.

**Project administration:** T. P. A. U. Thennakoon, S. Ekanayake.

**Supervision:** S. Ekanayake.

**Validation:** S. Ekanayake.

**Writing – original draft:** T. P. A. U. Thennakoon.

**Writing – review & editing:** S. Ekanayake.

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
