## [Decision Letter · Decision Letter 0]

18 Jul 2022

PONE-D-22-14882Full title: Sri Lankan traditional parboiled rice: a panacea for hyperglycaemia?PLOS ONE

Dear Dr. Ekanayake,

Thank you for submitting your manuscript to PLOS ONE. After careful consideration, we feel that it has merit but does not fully meet PLOS ONE’s publication criteria as it currently stands. Therefore, we invite you to submit a revised version of the manuscript that addresses the points raised during the review process.

We look forward to receiving your revised manuscript.

Kind regards,

Min Huang

Academic Editor

PLOS ONE

Journal Requirements:

Reviewers' comments:

Reviewer's Responses to Questions

**Comments to the Author**

1. Is the manuscript technically sound, and do the data support the conclusions?

Reviewer #1: Partly

Reviewer #2: Partly

2. Has the statistical analysis been performed appropriately and rigorously? 

Reviewer #1: Yes

Reviewer #2: Yes

3. Have the authors made all data underlying the findings in their manuscript fully available?

Reviewer #1: No

Reviewer #2: Yes

4. Is the manuscript presented in an intelligible fashion and written in standard English?

Reviewer #1: Yes

Reviewer #2: No

5. Review Comments to the Author

Reviewer #1: The study entitled 'Glycaemic response of Sri Lankan traditional parboiled rice' is a good study, which has been written well; however there are some concerns to address.

The comments are provided on the manuscript.

Good luck

Reviewer #2: 1. The subject matter of the MS is timely and relevant in view of the rice consumption and diabetes and obesity. The experimental part is voluminous as the data generated is on 6 cultivars and 3 processing variants. The GI is determined on mixed meal instead of single cereal namely the rice.

2. 100% brown rice and just 4% DM rice really doesnot make a significant difference, because, 4% DM rice also contains some amount of bran. It would have been better if they had polished the rice 8-10% which is the normal DM of the market samples.

3. Parboiling method is not a standard method, as usually, paddy is soaked in cold or hot water till it attains 25 - 35%, then subjected to live steam till the starch completely gelatinises and then dried. In the present study, the paddy is soaked in boiling water and heated till the grains were split and dried. The method may have also influenced the GI value.

The parboiled paddy was just dehusked (brown rice). It will have poor cooking quality as the bran hinders swelling of the grains during cooking. The authors should clarify these aspects.

4. The language of the MS is far from satisfactory wrt the use of appropriate scientific terminologies (eg L No. 23, Sri Lankan traditional rice

produce lower glycaemic responses, instead it should be Sri Lankan traditional rice elicit lower glycaemic responses, L. No. 147, varieties are stated in tables, it should be varieties are presnetdin tables ).

5. Digestible starch normally reduces on parboiling but in the case of

Dharnala sample it has increased ?

6. The total DF values are considerably on higher side?

7. The portion size of rice used for GI for parboiled rice is 1.5 to 2 times higher compared to raw as well as milled rice. How it is possible. This need satisfactory explaination

6. PLOS authors have the option to publish the peer review history of their article (what does this mean?). If published, this will include your full peer review and any attached files.

Reviewer #1: No

Reviewer #2: **Yes: **Dr. N. G. Malleshi

---

## [Author Response · Author response to Decision Letter 0]

25 Jul 2022

Editor comments - have been addressed in the Cover letter (in a table format) 

Reviewer comments - have been addressed in the response to reviewers document (in a table format)

---

## [Decision Letter · Decision Letter 1]

8 Aug 2022

Full title: Sri Lankan traditional parboiled rice: a panacea for hyperglycaemia?

PONE-D-22-14882R1

Dear Dr. Ekanayake,

We’re pleased to inform you that your manuscript has been judged scientifically suitable for publication and will be formally accepted for publication once it meets all outstanding technical requirements.

Kind regards,

Min Huang

Academic Editor

PLOS ONE

Reviewers' comments:

Reviewer's Responses to Questions

**Comments to the Author**

1. If the authors have adequately addressed your comments raised in a previous round of review and you feel that this manuscript is now acceptable for publication, you may indicate that here to bypass the “Comments to the Author” section, enter your conflict of interest statement in the “Confidential to Editor” section, and submit your "Accept" recommendation.

Reviewer #1: All comments have been addressed

2. Is the manuscript technically sound, and do the data support the conclusions?

Reviewer #1: Yes

3. Has the statistical analysis been performed appropriately and rigorously? 

Reviewer #1: Yes

4. Have the authors made all data underlying the findings in their manuscript fully available?

Reviewer #1: Yes

5. Is the manuscript presented in an intelligible fashion and written in standard English?

Reviewer #1: Yes

6. Review Comments to the Author

Reviewer #1: Rice is a major staple around the World, and it does contribute to the glycemic load. In the face of high global prevalence of type 2 diabetes and insulin resistance, healthy options of rice types need recognition.

This article is a step forward in the right direction.

Congratulations to the authors.

7. PLOS authors have the option to publish the peer review history of their article (what does this mean?). If published, this will include your full peer review and any attached files.

Reviewer #1: No

---

## [Editor Report · Acceptance letter]

4 Sep 2022

PONE-D-22-14882R1 

Sri Lankan traditional parboiled rice: a panacea for hyperglycaemia? 

Dear Dr. Ekanayake:

I'm pleased to inform you that your manuscript has been deemed suitable for publication in PLOS ONE. Congratulations! Your manuscript is now with our production department. 

Kind regards, 

on behalf of

Dr. Min Huang 

Academic Editor

PLOS ONE